# Effectiveness and Characterization of Severely Energy-Restricted Diets in People with Class III Obesity: Systematic Review and Meta-Analysis

**DOI:** 10.3390/bs9120144

**Published:** 2019-12-07

**Authors:** Gabrielle Maston, Alice A. Gibson, H. Reza Kahlaee, Janet Franklin, Elisa Manson, Amanda Sainsbury, Tania P. Markovic

**Affiliations:** 1The Boden Collaboration for Obesity, Nutrition, Exercise & Eating Disorders, Charles Perkins Centre, Faculty of Medicine and Health, The University of Sydney, Sydney 2006, Australia; alice.gibson@sydney.edu.au (A.A.G.); amanda.salis@sydney.edu.au (A.S.); tania.markovic@sydney.edu.au (T.P.M.); 2Metabolism & Obesity Services, Royal Prince Alfred Hospital, Sydney 2006, Australia; janet.franklin@health.nsw.gov.au (J.F.); elisia.manson@health.nsw.gov.au (E.M.); 3Faculty of Medicine and Health, School of Pharmacy, The University of Sydney, Sydney 2006, Australia; reza.kahlaee@sydney.edu.au; 4School of Life, and Environmental Sciences, Faculty of Science, The University of Sydney, Sydney 2006, Australia

**Keywords:** obesity, morbid, diet, reducing, meta-analysis, meta-regression, weight loss treatment, low-energy liquid diet

## Abstract

Severely energy-restricted diets are used in obesity management, but their efficacy in people with class III obesity (body mass index ≥40 kg/m^2^) is uncertain. The aims of this systematic review and meta-analysis were to determine the effectiveness and characteristics of severely energy-restricted diets in people with class III obesity. As there was a lack of publications reporting long-term dietary interventions and randomised controlled trial designs, our original publication inclusion criteria were broadened to include uncontrolled study designs and a higher upper limit of energy intake. Eligible publications reported studies including adults with class III obesity and that assessed a diet with daily energy intake ≤5000 kJ for ≥4 weeks. Among 572 unique publications from 4 databases, 11 were eligible and 10 were suitable for meta-analysis. Our original intention was to classify comparison arms into short-term (<6 months) and long-term (>1 year) interventions. Due to the lack of long-term data found, comparison arms were classified according to the commonalities in dietary intervention length among the included publications, namely dietary interventions of 4 weeks’ duration and those of ≥6 weeks’ duration. After a 4-week severely energy-restricted diet intervention, the pooled average weight loss was 9.81 (95% confidence interval 10.80, 8.83) kg, with a 95% prediction interval of 6.38 to 13.25 kg, representing a loss of approximately 4.1 to 8.6% of initial body weight. Diets ≥6 weeks’ duration produced 25.78 (29.42, 22.15) kg pooled average weight loss, with a 95% prediction interval of 13.77 to 37.80 kg, representing approximately 10.2 to 28.0% weight loss. Daily dietary prescriptions ranged from 330 to 5000 kJ (mean ± standard deviation 2260 ± 1400 kJ), and had wide variations in macronutrient composition. The diets were administered mostly via liquid meal replacement products. While the included publications had a moderate risk of bias score, which may inflate reported weight loss outcomes, the published data to date suggest that severely energy-restricted diets, delivered via diets of varying composition, effectively produce clinically relevant weight loss (≥10% of initial body weight) when used for 6 weeks or more in people with class III obesity.

## 1. Introduction

Obesity is a global issue affecting 2.1 billion people worldwide [1]. Its prevalence and severity continues to rise [1,2,3,4,5,6], thereby increasing the disease burden [7]. Co-morbid conditions and mortality increase progressively as body mass index (BMI) rises from 25 kg/m^2^ and above [8,9]. Individuals with BMI ≥ 40 kg/m^2^—defined as class III obesity by the World Health Organisation (WHO) [10]—have been shown to have an 80% increase in hazard ratios for all-cause mortality compared to those with BMI < 25 kg/m^2^ [9]. Intentional weight losses from lifestyle-based interventions in people with overweight and obesity reduce metabolic risk factors for the development of co-morbid conditions and decreases all-cause mortality by 25% [8,11,12,13,14,15]. Therefore, intentional weight loss therapy is an important means of preventing obesity-related disease and early death.

Commonly-used and effective long-term weight management treatments for class III obesity are bariatric surgery and pharmacotherapy [16,17,18,19,20]; however, these treatments may be medically contraindicated for some individuals with class III obesity due to anesthetic risk and negative interactions with other co-morbid conditions [20,21], and can also be financially and geographically inaccessible for many people [19,22,23,24,25]. A promising alternative to these treatments is severely energy-restricted diets, a broad defining term that includes very low energy diets (VLEDs) and low energy diets (LEDs). VLEDs are defined by The United Nations food standards body Codex Alimentarius as restricting daily energy intake to 2500 to 3300 kJ (600 to 800 kcal), and LEDs are defined as restricting daily energy intake to between 4200 kJ (1000 kcal) and 5000 kJ (1200 kcal) [10,26,27]. VLEDs are commonly implemented with the use of total meal replacement (TMR) diets, which replace the majority of meals and snacks with formula foods such as shakes, soups, bars and desserts, and sometimes allow selected low-energy additional items for consumption, such as tea, coffee and low-starch vegetables. LEDs are commonly administered in partial meal replacement diets, involving the partial use of meal replacement products and prescribed food. These diets can result in weight losses of 7 to 17 kg (6 to 16% of initial body weight) at 1 year in cohorts of people with overweight or obesity (BMI in participants ranging from 25 to 42 kg/m^2^) [28,29,30,31,32].

Australian clinical practice guidelines for the management of overweight and obesity state that severely energy-restricted TMR diets may be a suitable option when used in a medically supervised weight loss program for individuals with a BMI ≥ 30 kg/m^2^, or individuals with a BMI ≥ 27 kg/m^2^ and obesity-related comorbidities requiring rapid weight loss [33]. Such diets are not recommended in the routine management of obesity in the National Institute for Health and Care Excellence (NICE) clinical guidelines from the United Kingdom (UK), nor in the American College of Cardiology Task Force clinical guidelines from the United States of America (USA) [34,35]. The UK guidelines suggest that severely energy-restricted diets are ineffective in the long-term, are nutritionally unbalanced, may be harmful, and should only be considered as part of a multi-component weight management strategy for people with obesity who have a clinical need for rapid weight loss [34]. Similarly, the USA guidelines state that there is insufficient evidence to support the use of severely energy-restricted diets because long-term evidence on weight loss maintenance is lacking, and that such diets should only be used in limited circumstances under the supervision of trained healthcare providers [35]. In contrast, the position statement from the Academy of Nutrition and Dietetics in the USA supports the use of LEDs, administered as meal replacement diets, if the diet forms part of a comprehensive weight loss and weight maintenance program [36]. LEDs are considered a suitable weight management option as they provide portion control, dietary structure and a significant reduction in energy intake. However, the position statement does not support the use of VLEDs, citing poor long-term weight loss maintenance as a reason [36]. The aforementioned guidelines are not only at odds with each other, but also with previous systematic reviews and meta-analyses that showed severely energy-restricted diets are safe and effective for long-term weight management (≥ 1 year) and treatment of obesity-related comorbidities for people with a BMI ≥ 25 kg/m^2^ [29,37,38].

In our clinical experience, individuals with class III obesity, compared to those with class I obesity (≥30 to <35 kg/m^2^) or class II obesity (≥35 to <40 kg/m^2^), have higher dropout rates from dietary obesity treatment programs, greater challenges with diet adherence associated with hedonic eating, more mental health issues, greater complexity in their social situations, more financial instability, and a greater number of medical conditions, all of which seem to hinder their success on a severely energy-restricted diet. Our clinical observations are supported by current literature showing that medical and social challenges worsen with increasing BMI [21,39,40,41,42]. For instance, as BMI increases there is a higher prevalence of binge eating [43,44], social discrimination [39] and other co-morbid conditions [8,9], all of which are known contributors to co-morbid psychological conditions such as anxiety and depression [45,46,47,48,49]. The presence of any one of these factors strongly predicts poor adherence to dietary and lifestyle treatments for obesity [50,51,52]. Moreover, social discrimination affects the ability to find employment, contributing to financial instability [53,54]. Individuals with obesity who are unemployed are 31 times more likely to drop out of a dietary/lifestyle obesity treatment program than those who are employed [55]. We hypothesise that due to these barriers, individuals with class III obesity may not achieve clinically significant weight loss when undergoing a severely energy-restricted diet program.

A search of Google Scholar and the electronic databases Cochrane and EMBASE with the terms ‘systematic review’ and synonyms for class III obesity identified no systematic reviews of studies of severely energy-restricted diets in people with class III obesity, so the usefulness of these diets in this population is unclear. Thus, the primary aim of this investigation was to determine the effectiveness of severely energy-restricted diets in individuals with class III obesity. The secondary aim was to qualitatively review the way in which these diets were used in this group.

## 2. Materials and Methods

This review is reported in accordance with the Preferred Reporting Items for Systematic Reviews and Meta-Analyses (PRISMA) guidelines. The review was prospectively registered with PROSPERO (CRD42017058897) on 26 March 2017. Due to the limited number of randomised controlled trials and total publications initially retrieved using the prospectively-registered inclusion criteria, the inclusion criteria were broadened to include publications in which the interventions were uncontrolled, or which reported on prospective or retrospective case series or cohorts, as well as those with dietary prescriptions with a higher upper limit of energy intake. In addition, due to the lack of publications reporting weight loss interventions of ≥1 year in duration, as per our original inclusion criteria, dietary interventions were categorised using shorter dietary intervention durations, i.e., 4 weeks’ duration and ≥6 weeks’ duration.

### 2.1. Search Strategy

An electronic literature search was performed with no language or beginning date restriction until June 2019 using the following databases: MEDLINE, Cochrane CENTRAL, Embase, and Scopus. Our search used the following Boolean keyword combination: (i) ‘obesity, morbid’; ‘super obese’; ‘severe obese’ or ‘extreme obese’ and (ii) ‘VLED’; ‘VLCD’; ‘very low energy diet’; ‘very low calorie diet’; ‘LED’; ‘low energy diet’; ‘LCD’; ‘low calorie diet’; ‘diet therapy’; ‘caloric restriction’; ‘diet carbohydrate-restricted’; ‘diet fat-restricted’ or ‘ketogenic diet’. No restrictions were put in place regarding diet duration or reporting timeframe. The reference lists of publications included in this review were searched for additional publications that may have been missed in our search strategy.

### 2.2. Publication Eligibility Criteria

Publications needed to report investigations including adults with class III obesity with or without obesity-related co-morbidities with a diet prescribing a maximum daily energy intake of 5000 kJ (1200 kcal) for at least 4 weeks in at least one comparison arm. The threshold of <5000 kJ (1200 kcal) per day is considered a severe dietary energy restriction for people with class III obesity because it represents at least a 65% restriction relative to estimated total energy expenditure [56]. This energy intake threshold allowed the inclusion of both VLEDs and LEDs. VLEDs and LEDs can be achieved with (i) a TMR diet, (ii) a complete food-based (FB) diet, or (iii) a combination of the two (i.e., a partial meal replacement diet) [57]. Our dietary inclusion criteria allowed for all three forms of dietary prescription.

Publications reporting interventions that induced energy restriction through a surgical procedure or implanted device to create appetite suppression or physical restriction to food or liquid consumption during the dietary intervention in all comparison arms were excluded. Publications in which pharmacological weight loss agents were used in all comparison arms during the period of severe dietary energy restriction were excluded, unless their use was after the severely energy-restricted diet (e.g., during the weight maintenance phase of an intervention). Publications containing a behavioral or lifestyle therapy intervention component were included. Only full papers were considered for inclusion, but they could involve reporting of either a prospective or retrospective cohort. Eleven publications were identified for data extraction. Figure 1 shows an overview of the selection process.

### 2.3. Data Extraction

Data about the participants, interventions and results were extracted using a piloted data collection form. One author (G.M.) extracted all the relevant data from the publications. A second author (A.A.G.) independently reviewed publications during the title and abstract screening process and checked the data extracted. Authors of the publications included in this review were contacted for any missing data.

### 2.4. Data Cleaning

Four publications contained more than one intervention arm [58,59,60,61], and all intervention arms were eligible for inclusion as separate comparison arms. Due to the limited number of publications found reporting long-term dietary interventions, comparison arms from each publication were separated and categorised based on the dietary intervention durations of 4 weeks and ≥6 weeks. There were no interventions of 5 weeks’ duration. The energy intake prescription of dietary obesity treatments reported in kilocalories (kcal) were converted to kilojoules (kJ) by multiplying by 4.18. For dietary interventions with durations reported in months or years, duration was converted to weeks, assuming 4.3 weeks per month. The interventions were further classified as being delivered via a TMR, a FB diet, or a partial meal replacement diet. TMR diets were defined as replacing all meals and snacks with formula foods such as shakes, soups, bars and desserts, and sometimes allowing selected additional low-energy items for consumption such as tea, coffee and low-starch vegetables. FB diets were defined as a prescribed diet consisting of a combination of different food and drinks, without the use of meal replacement products. The presence or absence of adjunct therapies (i.e., behavioural therapy or pharmacotherapy) and their duration, were also extracted. While our prospectively-registered PROSPERO protocol (CRD42017058897) stated that data pertaining to body composition would be collected and reported, no such data was found in the retrieved publications.

### 2.5. Data Analysis

#### 2.5.1. Primary Outcomes

Our primary outcome was the mean difference in weight (kg) from baseline (prior to the dietary treatment) to completion of the study (inclusive of the severely energy-restricted diet and any follow-up time, which may have included a weight maintenance diet).

#### 2.5.2. Secondary Outcomes

Our secondary outcomes were the energy content and macronutrient composition of the prescribed dietary protocols used during the severely energy-restricted diet, the use of any meal replacement products or foods, and details of any adjunct therapies used with the intention of maximising the effectiveness of the primary dietary obesity treatment.

#### 2.5.3. Meta-Analysis and Meta-Regression

The meta-analysis was conducted using a random effects model. Pooled results were calculated using the weighted mean difference in kg, with a 95% confidence interval (CI). In one publication that did not report mean weight loss [61], mean end weight was subtracted from mean baseline weight, and a standard deviation of the change in weight was calculated using the reported *p*-value for the mean change in weight, because the result for a Student’s *t*-test was quoted in that publication. The *t*-value was obtained from a Student’s *t* distribution table, and the sample size (n) was then used to calculate the degrees of freedom (df, where df = n−1). We then calculated the standard deviation of change (SD_c_) in weight using the equation SD_c_ = tinv (*p*-value, df), where tinv refers to a two-tailed Student’s *t* distribution, as per Cochrane guidelines [62].

The second method used to calculate standard deviation for mean weight loss for another publication [60] which provided sample size, median and range was the Hozo method [63]. The Hozo method provides an estimate of standard deviation of the change, using the following equation.
(1)$${\mathrm{SD}}_{c}\;=\;\surd \;\left[\frac{1}{12\;}\left\{\;\frac{{\left(a-2m+b\right)}^{2}}{4}+\;{\left(b-4\right)}^{2}\;\right\}\right]$$
where, SD_c_ = standard deviation of the change between baseline and end weight.
a = change between baseline and end weight, low end of the range of the sample.b = change between baseline and end weight, high end of the range of the sample.m = change between baseline and end weight median.

The third method used to calculate the standard deviation of the change in weight for the remaining publication [64], when the aforementioned methods could not be used, was the imputation method, when the standard deviation was provided for baseline and end weight means, as outlined in the Cochrane Handbook and shown in the following formula [62].
SD_c_ = √[SD^2^_b_ + SD^2^_f_ – (2 × Corr × SD_b_ × SD_f_)](2)
where, SD_c_ = standard deviation of the change between baseline and end weight.
SD_b_ = standard deviation of baseline weight mean.SD_f_ = standard deviation of end weight mean.Corr = Correlation coefficient between the ‘baseline’ standard deviation and the ‘end’ standard deviation.

The correlation coefficient was calculated from the only publication included in the meta-analysis [61] that reported the standard deviation of baseline and end weight mean, and in which the standard deviation for change in weight could be calculated using the reported *p*-value.

The I-squared (I^2^) statistic, representing the proportion of the observed heterogeneity attributable to underlying heterogeneity, was used to quantify statistical heterogenicity. To explore the potential covariate factors affecting statistical heterogeneity in our meta-analyses, a conventional meta–regression was performed using the I^2^ statistic (≥50%), chi–squared (*p* < 0.05), and the estimate of between-publication variance tau-squared (τ^2^). The meta-regression included the use of publication-level covariates to provide effect-estimates (ESs) on the publications used in the meta-analysis. Random-effects univariable and multivariable meta-regression analyses, using the restricted maximum likelihood (REML) method [65], were conducted on the following covariates: type of study design, percentage of female participants, percentage of participants completing the study, mean age, energy prescription consistent or changing over time, and daily energy intake prescription above or below 2300 kJ (550 kcal). An adjusted R^2^ statistic (aR^2^) was calculated to determine the proportion of between-publication variance explained by individual covariates. The meta regression was performed using standard protocols [66]. All statistical analyses were conducted using STATA/IC version 14.2, Windows 64 bit (Stata Corp LP, College Station, TX, USA).

### 2.6. Risk of Bias and Quality Assessment

One author (G.M.) assessed the risk of bias of all 11 publications included in the systematic review using the ROBINS-1 tool [67], which assesses bias according to the following domains: potential for systematic errors causing confounding, selection of participants into the intervention, classification of interventions due to prior knowledge of the expected outcome, suspected deviations from intended interventions such as adherence to the assigned intervention, suspected missing data, measurement of outcomes due to non-blinding of investigators and participants, and suspected selective reporting of the results. Publications could be scored in each domain as having unclear, low, moderate, high (serious) or extreme (critical) risk of bias, using published criteria [67]. After collating the results of individual domains, an overall score was determined using three categories: low, moderate or high risk of bias, again based on published guidelines [67]. For example, to achieve an overall low risk of bias score, all seven domains must be scored as low risk of bias. Moderate risk of bias score is achieved when all seven domains are scored with a combination of low or moderate risk of bias. In contrast, the achievement of one high risk of bias score in a standalone domain is carried forward to the overall risk of bias score, unless superseded by a score of extreme risk of bias in at least one domain.

## 3. Results

### 3.1. Publication Selection

Our search strategy identified 951 publications. EndNote™ reference management software was used to identify and remove duplicate publications resulting in 572 unique publications. The titles and abstracts of these 572 publications were screened according to our selection criteria, resulting in the retention of 157 publications. Full texts of these 157 publications were accessed and screened, resulting in 146 publications being excluded (see Figure 1 for details). Eleven publications met our selection criteria [58,59,60,61,64,68,69,70,71,72,73], of which ten publications—which included 15 comparison arms—were able to be used in our meta-analyses [58,59,60,61,64,68,69,70,71,72]. One publication [73], which was included in the systematic review, had to be excluded from the meta-analyses as it did not report weight in absolute terms nor provide a final measurement for average weight, from which weight loss could be calculated. Attempts to contact the author of the publication to obtain missing data were unsuccessful.

### 3.2. Risk of Bias and Quality Assessment

Nine of the 11 (82%) publications in this review achieved an overall bias score of moderate risk, one had a low overall risk, and another a high risk of bias score (Table 1). The areas in which bias was mainly found were (i) suspected confounding (n = 10 publications, 91%) and this was largely from co-morbid conditions and baseline weight, (ii) selection of participants into the intervention (n = 9, 82%), and (iii) classification of interventions (n = 6, 55%).

In four publications (four comparison arms) [64,68,70,71], only participants who had successfully completed a TMR diet program and had lost a targeted amount of weight were selected, resulting in artificially high participant diet completion rates (83 to 100%). Ten publications used non-randomised methods for participant allocation into intervention arms [58,59,60,61,64,68,69,70,71,73], and in six of these the classification of the intervention status was potentially affected by knowledge of the outcome by the investigators, as the intervention status was unclear at baseline, classified retrospectively, or prior to the point of delivery of the intervention [59,61,64,68,70,71].

### 3.3. Characteristics of Publications Included in the Meta-Analysis

The characteristics of the 11 publications in our systematic review (including the 10 in our meta-analyses) are shown in Table 2. Two publications were conducted in an inpatient setting [58,59], and the remaining nine in an outpatient environment [60,61,64,68,69,70,71,72,73].

Of the ten publications included in meta-analysis, three [58,59,61] contained more than one comparison arm, resulting in eight comparison arms in these three publications. The eight comparison arms were reviewed to ensure there was no duplication of population data. Each of the eight comparison arms had a different dietary protocol and used a different group of participants, and were thus included as separate comparators. This resulted in a total of 15 comparison arms for the meta-analyses. The meta-analysis of severely energy-restricted diets of 4 weeks’ duration contained nine comparison arms from four publications [58,59,61,70], which were publications reporting on prospective case series [58,59] and prospective cohorts [61,70]. The meta-analysis of severely energy-restricted diets ≥6 weeks’ duration contained six comparison arms from six publications [60,64,68,69,71], which reported on non-randomised controlled trials [60,71], prospective case series [64,69], a retrospective case series [68] and a randomised controlled trial [72].

### 3.4. Participants

Of the ten publications (with 15 comparison arms) identified for use in our meta-analyses, three publications excluded participants with major psychiatric disorders [60,69,72], three excluded conditions that contraindicate the use of severely energy-restricted diets, namely cardiac, renal and liver failure [60,69,72], and one excluded subjects with irregular life habits, namely addictive behaviors and sleep problems [71]. With respect to obesity-related conditions, four publications included participants with type 2 diabetes mellitus [60,61,68,70], three included participants with hypertension [60,68,72] and one included participants with dyslipidemia [68]. The two remaining publications did not mention exclusion or inclusion criteria; however, they reported that participants had no medical concerns [58,59].

The mean age ± SD of participants from the ten publications included in the meta-analyses was 42.9 ± 10.1 years, and mean BMI was 48.6 ± 5.3 kg/m^2^. Similar results were seen for the 11 publications (16 comparison arms) included in the systematic review, where the mean age of participants was 43.2 ± 9.8 years, and mean BMI was 43.2 ± 9.8 kg/m^2^. Participant details of each comparison arm are shown in Table 2.

#### 3.4.1. Participant Data for the Meta-Analysis of Interventions with 4-Week Duration

The sample size of each of the nine comparison arms in these four publications [58,59,61,70] ranged from 5 to 48, giving a total of 166 participants in the meta-analysis of severely energy-restricted diets with 4-week duration. In all of these publications there was a greater proportion of women, ranging from 60 to 100%.

#### 3.4.2. Participant Data for the Meta-Analysis of Interventions ≥6 Weeks in Duration

The sample size of each of the six comparison arms from these six publications ranged from 5 to 119, giving a total of 171 subjects for inclusion in the meta-analysis of dietary interventions ≥6 weeks’ duration. Three publications had a greater proportion of women, ranging from 59.0 to 83.5% [60,69,72], and the other three had more men, ranging from 57 to 83% [64,68,71].

### 3.5. Meta-Analysis of Interventions with 4-Week Duration

The meta-analysis of severely energy-restricted diets of 4 weeks’ duration (Figure 2) showed a pooled average weight loss of 9.81 (95% confidence interval 10.80, 8.83) kg [58,59,61,70]. This represents an approximately 8.3% loss of initial body weight. The 95% prediction interval for this subgroup showed that weight losses of 6.38 to 13.25 kg would be expected from future treatment settings similar to those used in the comparison arms in this meta-analysis. This prediction interval represents a loss of approximately 4.1 to 8.6% of initial body weight.

One of the nine comparison arms was identified as an outlier because while the intervention was 4 weeks’ duration like the other’s comparison arms, it had a reporting time of 30 weeks [61], whereas reporting time was only 4 weeks in the other eight comparison arms [58,59,61,70]. After excluding this outlier, the results of our meta-analysis were similar, with a pooled average weight loss of 10.22 (11.24, 9.20) kg.

### 3.6. Meta-Analysis of Interventions ≥6 Weeks in Duration

The meta-analysis of severely energy-restricted diets of ≥6 weeks duration (median intervention length 10 weeks) showed a pooled average weight loss of 25.78 (29.42, 22.15) kg (Figure 2) [60,64,68,69,71]. This represents an approximately 17.2% loss of initial body weight. The 95% prediction interval for this subgroup showed that weight losses of 13.77 to 37.80 kg would be expected to be observed in similar future treatment settings to those used in the comparison arms in this meta-analysis. This prediction interval represents approximately 10.2 to 28.0% loss of initial body weight.

As this meta-analysis included data from one randomised controlled trial, with the remaining data from uncontrolled interventions, the controlled data [72] was removed to determine if this skewed the results of the analysis. However, a similar result was found after its removal, with a pooled average weight loss of 26.90 (29.44, 24.40) kg.

One publication [68] was identified as an outlier because it had a serious risk of bias (Table 1), and also reported a longer dietary intervention (142.0 ± 43.0 weeks) compared to the other comparison arms that had dietary interventions with mean and median durations of 35.8 and 12.0 weeks, respectively (Table 2). However, when removing this publication from our meta-analysis, the results of our meta-analysis were similar, with a pooled average weight loss of 24.51 (28.26, 20.76) kg. Due to the limited effect of this outlier on our results, the outlier was included in our final reported results and conclusions, as cited here and in the abstract.

### 3.7. Meta-Regression of Factors Affecting Heterogeneity

The measure of between-publication heterogeneity for all 15 comparison arms was represented by the I^2^ statistic (99.9%, Figure 2) and τ^2^ (0.2), both of which were significant. The estimate of between-publication variance, τ^2^, is the standard deviation of true effect size, which is the difference in the effect sizes for each publication [66]. Due to the high heterogeneity observed, a univariable and multivariable meta–regression was performed. The results of our univariable and multivariable meta-regression included the use of covariate factors to provide effect-estimates (ESs) on the ten publications used in the meta-analysis. Using the aR^2^ statistic to determine the proportion of between-publication variance explained by individual covariate factors, three publication-level covariate factors (type of publication design, % female and % completers) were found to be significant contributors to heterogeneity. A multivariable meta–regression analysis was also conducted, and confirmed this result (aR^2^ = 88.08%). While this analysis cannot determine to which extent or the direction that covariates affect heterogeneity, nor the degree of weight loss shown, it can provide an indication of why high heterogeneity may be occurring (Figure 3). The risk of bias assessment identified four publications (with four comparison arms) [64,68,70,71] that had artificially high participant dietary completion rates due to suspected biases in publication design. To determine if the potential source of bias had affected identification of the covariate factor ‘% completers’ as a source of heterogeneity, the meta-regression was performed a second time. Comparison arms that were identified as having a high or moderate risk of bias score and which obtained 100% participant dietary completion rates were removed. After removing the four comparison arms, there was no change in the outcomes of the meta-regression.

### 3.8. Publication Not Included in Meta-Analysis

The weight loss outcome from the one comparison arm pertaining to the single publication excluded from meta-analysis [73] was consistent with the results of the meta-analysis. This comparison arm [73] was a 12-week severely energy-restricted TMR intervention with the addition of low starch vegetables (total daily energy prescription of 3344 kJ (800 kcal), 100 g carbohydrate, 15 g fat, 70 g protein) and behavioural therapy which resulted in an 18% (approximately 24 kg) mean reduction in body weight. This result is similar to that from our meta-analysis, where a pooled average weight loss of 25.78 (29.42, 22.15) kg (approximately 17.2%) was achieved with severely energy-restricted diets of ≥6 weeks with no adjunct treatment (Figure 2).

### 3.9. Severely Energy-Restricted Dietary Protocols

The daily energy content of the severely energy-restricted diets used in the 16 comparison arms of the 11 publications included in our systematic review ranged from 330 kJ (80 kcal) to 5000 kJ (1200 kcal), with a mean ± SD of 2257 ± 1402 kJ (540 ± 335 kcal) per day. Twelve of the comparison arms could be classified as VLEDs, with a mean daily energy intake prescription of 1661 ± 1010 kJ (399 ± 242 kcal) [58,59,60,61,64,70,71,73]. The remaining four comparison arms could be classified as LEDs, with a daily energy intake prescription of between 3344 kJ (800 kcal) and 5000 kJ (1200 kcal) [61,68,69,72]. The comparison arm with the highest prescribed daily energy intake 5000 kJ (1200 kcal) for 4 weeks—resulted in an average weight loss of 7.00 (7.44, 6.56) kg at the 30-week reporting time [61], which is similar to the pooled average weight loss of 9.81 kg in the meta-analysis of 4-week severely energy-restricted diets (Figure 2). There was also variability in the macronutrient profiles in all 16 comparison arms of these diets (Table 2). The daily protein prescription ranged from 17 to 90 g (mean 55 ± 27 g), and the daily carbohydrate prescription ranged from 0 to 115 g (mean 47 ± 47 g).

Neither the energy prescription nor the macronutrient profile appeared to affect weight loss outcomes. For example, an arm of a 4-week intervention [58] with an initial prescribed daily energy intake of 330 kJ (80 kcal), increasing to 740 kJ (176 kcal) after 2 weeks, a macronutrient profile of 17 g protein daily and a varying daily carbohydrate content of 0 g for 2 weeks increasing to 25.5 g thereafter, produced an average weight loss of 11.20 (12.25, 10.15) kg. Another comparison arm [61] using a dietary protocol with a higher daily energy, protein and carbohydrate prescription resulted in a similar average weight loss of 12.80 (18.72, 6.88) kg. Similar findings from other publications (Table 2) indicate that there may be wide-ranging macronutrient profiles that can be used during severely energy-restricted diets without affecting weight loss outcomes.

Fourteen of the 16 comparison arms (10 of 11 publications) used a severely energy-restricted diet protocol consisting of a TMR diet, with variable use of additional food and fluids [58,59,60,64,68,69,70,71,72,73]. Two comparison arms used a TMR diet with low-starch vegetables [60,73], six allowed consumption of black tea and coffee [58,59], and one allowed up to 750 mL diet soft drink daily [59]. Of the 16 comparison arms in this review, only two from one publication [61] achieved severe energy restriction through the use of food, but only in the last 20 days of a 30-day intervention, after using a TMR diet for the initial 10 days.

The two forms of adjunct therapy, behavioural therapy and pharmacotherapy, were used in seven comparison arms. Behavioural therapy was used in seven of 16 comparison arms [60,64,69,70,71,72,73], and pharmacotherapy was used during the weight maintenance phase in one publication [72] in which sibutramine, orlistat or diethlypropion hydrochloride were used for an unknown length of time. The outcomes were similar to the remaining nine comparison arms not using any type of adjunct therapy [58,59,61,68]. For example, a mean weight loss of 17.10 (19.87, 14.33) kg was achieved in a 4-week comparison arm in which behavioural therapy was used [70], while 12.80 (18.72, 6.88) kg weight loss was achieved in another 4-week comparison arm in which there was no adjunctive treatment [61]. There was a lack of detail provided regarding the behavioural and pharmacotherapeutic treatments used, therefore it remains to be determined whether these forms of adjunctive treatment are useful in this setting.

## 4. Discussion

In this investigation, publications reporting on the use of severely energy-restricted diets in individuals with class III obesity were systematically collated and reviewed. The meta-analysis demonstrated that a 4-week severely energy-restricted diet produced a pooled average weight loss of 9.81 kg or approximately 8.3% body weight reduction in this population. The prediction interval shown in the meta-analysis shows the expected weight loss to be between 6.38 and 13.25 kg. This would be below the 10% weight loss threshold for clinical significance for most people with class III obesity, depending on baseline weight. In contrast, interventions involving ≥6 weeks of a severely energy-restricted diet produced a pooled average weight loss of 25.78 kg, equating to approximately 17.2% reduction in body weight, depending on baseline weight. The prediction interval shows an expected weight loss of between 13.77 and 37.80 kg (which would represent 10.2 to 28.0% of initial body weight for most people with class III obesity), which is greater than the 10% threshold for clinically significant weight loss. Larger weight losses can potentially be achieved with extended dietary intervention lengths well beyond 6 weeks, as observed in a severely energy-restricted diet of approximately 142 weeks’ duration resulting in a mean weight loss of 44 kg (27% of initial body weight) [68].

Clinical practice guidelines in Australia and the USA suggest that a 10% reduction in body weight is a suitable target for people with class III obesity to decrease health risk and improve markers of cardiometabolic health [33,74], whereas UK guidelines suggest individuals with overweight and obesity should aim to lose as much weight as possible, with the realistic weight loss target of 5% or more of their initial body weight, advising that greater weight loss augments health benefits [75]. This is supported by recent evidence that the health benefits of weight loss are dose-dependent, with greater weight loss resulting in more health benefits [76,77,78,79,80,81]. The current investigation extends previous findings by demonstrating that individuals with class III obesity, using severely energy-restricted diets, can achieve similar weight loss to that previously shown with such diets in mixed cohorts of individuals with overweight and obesity (7 to 17 kg or 6 to 16% of initial body weight) [28,29,30,31,32]. This finding is contrary to our initial hypothesis, based on clinical observations and theoretical considerations (such as increased hedonic eating as well as medical and socioeconomic factors) that weight loss would be harder to achieve in people with class III obesity.

The analysis of dietary protocols used to achieve clinically significant weight loss in this review showed a wide variation in dietary protocols, suggesting that there may be a wide variation in energy prescription and macronutrient profiles that can be used effectively during a severely energy-restricted diet intervention. In ten publications, severely energy-restricted TMR diets were used, with only four of these ten publications allowing the use of supplemental food and beverages. While behavioural therapy was used in seven publications (and 7 of 16 comparison arms), weight loss was no greater in these publications/comparison arms, suggesting that behavioural therapy may not be necessary for clinically significant weight loss, at least up to 24 weeks, the maximum length of the seven comparison arms in this population. It has previously been found that adjunct therapies, such as behavioural therapy and pharmacotherapy, may improve weight loss outcomes during a severely energy-restricted TMR intervention [32,38,82,83] that had a follow up duration of 1 year. A previous meta-analysis showed that when a severely energy-restricted dietary protocol is combined with behaviour therapy, it produces an additional 3.9 kg of weight loss at 1 year compared to food-based diets or interventions using lifestyle behaviour therapy alone (without behavioural therapy) in mixed population cohorts with overweight and obesity (BMI ≥ 25 kg/m^2^) [37] and contributes to greater dietary compliance [32]. However, the effectiveness of behavioural therapy and pharmacotherapy as adjunctive therapies could not be fully assessed in the current review because of the limited number of publications available for analysis, lack of detail provided regarding treatment protocols, and minimal long-term data when these treatments may have their maximal beneficial effects. Thus, further trials are needed to determine the usefulness of behavioural therapy and pharmacotherapy as an adjunct to severely energy-restricted diets in people with class III obesity.

Poor compliance to a severely energy-restricted TMR diet is commonly noted by healthcare professionals and is often a criticism against their use in clinical practice [84]. However, the literature suggests that compliance to a severely energy-restricted TMR diet is higher than food-based diets [85,86], with one study showing 21.7% of participants completing a TMR diet intervention and achieving ≥ 10% weight loss, compared to 0% of participants in food-based control diet intervention [85]. Compliance with a structured and multi-faceted intervention involving a severely energy-restricted TMR diet has also been previously measured through clinic attendance records among a mixed cohort of 308 participants with overweight and obesity. Low compliance was found in 20% of participants, whereas medium and high compliance accounted for the remaining 80%. Of those who achieved medium and high compliance, 83% and 96%, respectively, achieved a clinically significant weight loss of >10% of initial body weight [87]. Concerns regarding compliance expressed by healthcare professions may vary according to the structure of the severely energy-restricted diet program used, different clinical contexts, and the expected weight loss targets and outcomes. Strategies that have been shown to improve compliance during a severely energy-restricted TMR diet in mixed population groups with overweight and obesity include the use of adjunctive therapies such as behavioural therapy to address poor lifestyle behaviours [38] via e.g. encouragement to self-monitor dietary intake [88], pharmacotherapy to reduce the drive to eat [83], tailoring dietary interventions to meet dietary preferences and nutrition requirements [88], and regular follow up with a healthcare professional [38].

This systematic review had a number of limitations and strengths. Most notably among limitations was the fact that the majority of publications included were identified as having a moderate risk of bias and there was a lack of randomised controlled trials, such that the weight loss achieved may be an overestimate. High quality randomised controlled trials are needed to confirm the current findings; however, randomised controlled trials are uncommon in the treatment of overweight and obesity. This is due to ethical constraints in withholding treatment from control groups, when treatments are known to improve health outcomes. Another limitation was the lack of data found in the identified publications. This lack of reported data may add a degree of measurement error, as statistical calculations require mathematical assumptions and best estimates. Therefore, the results of our meta-analysis should be viewed with a degree of caution. Many of the publications excluded people with eating disorders or addictive behaviors, which more commonly affect those with more severe degrees of obesity, so findings may not be applicable to people with these conditions. Many of the comparison arms reported weight loss outcomes immediately or shortly after completion of the dietary intervention, but it is well known that compliance to any weight loss/maintenance intervention wanes over time [50] and that weight regain is common [89]. Our meta-analyses do not address successful weight loss maintenance, which is defined as intentionally attaining and maintaining a body weight that is 10% or less than initial body weight for ≥1 year [89]. The effectiveness severely energy-restricted diets during weight loss maintenance for ≥1 year has previously been explored and it has been shown that there is large variability in the percentage of initial weight loss regain, which can range from −7 to 122% at 1 year to 26 to 121% at 5 years in mixed populations with overweight and obesity [38]. Future trials with follow up of at least 1 year are needed to determine durability of the diet-induced weight loss in populations with class III obesity (BMI ≥ 40 kg/m^2^). However, a strength of our investigation was the ability to group publication comparison arms into two separate subgroups for meta-analysis according to relatively unified diet durations, reducing the potential for variability in this domain. In addition, half of the publications included participants with cardio-metabolic conditions, including type 2 diabetes mellitus and hypertension, improving the generalisability of these findings to those with these common obesity-related conditions.

The findings from our review may be of significant interest to healthcare professionals, as we show that severely energy-restricted diets of ≥6 weeks’ duration, as opposed to those of ≤4 weeks’ duration, can achieve clinically relevant weight losses of 10% or more in people with class III obesity. Furthermore, a wide variation in dietary prescription can be used, with or without adjunctive therapy or pharmacotherapy, if the diet prescribed delivers between 330 kJ (80 kcal) to 5000 kJ (1200 kcal) per day. This provides healthcare professionals some clarity on expected weight loss outcomes in people with class III obesity, and general guidance on how severely energy-restricted diets can be prescribed in clinical practice.

## 5. Conclusions

While the results of this review should be interpreted with caution due to lack of controlled trials used in the meta-analysis, severely energy-restricted diets, most commonly achieved with TMR diets, may be an effective way for people with class III obesity to reduce their weight. Weight loss with these diets may be achieved with an average daily energy intake prescription of approximately 2300 kJ (540 kcal), ranging from 330 to 5000 kJ (80 to 1200 kcal), with a wide range of macronutrient profiles, diet durations, with or without food-based extras, behavioural therapy or pharmacotherapy. To achieve clinically significant weight losses (i.e., ≥10% of initial body weight), a diet duration of ≥6 weeks should be implemented, as this results in a predicted weight loss of approximately 14 to 38 kg (10 to 28% of initial body weight, depending on starting weight). Severely energy-restricted diets should be considered a viable form of weight management therapy for people with class III obesity, especially given that other obesity treatments, notably pharmacotherapy and surgery, may be contraindicated or unavailable.

## Figures and Tables

**Figure 1 behavsci-09-00144-f001:**
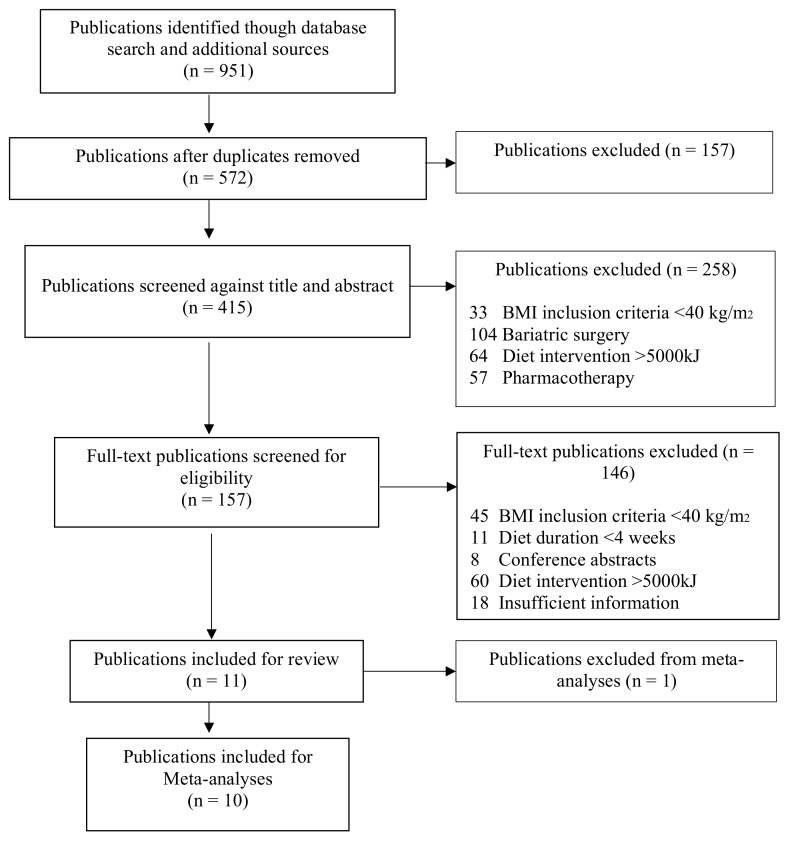
Overview of systematic search process and publication inclusion.

**Figure 2 behavsci-09-00144-f002:**
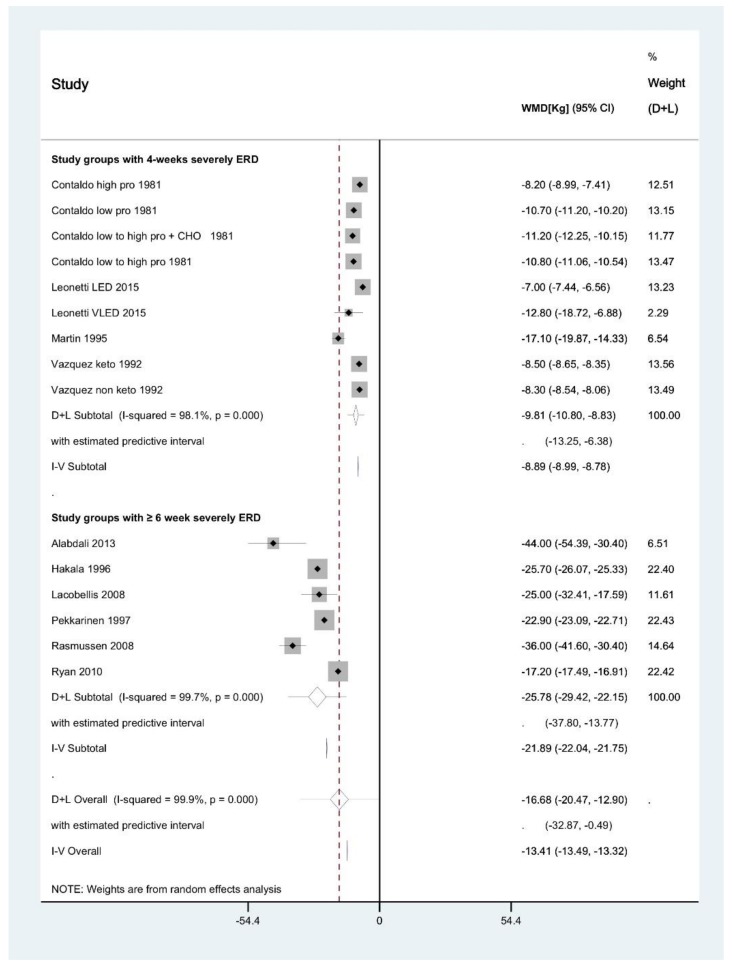
Meta-analysis of severely energy-restricted diets of 4 weeks or ≥6 weeks in duration. Meta-analysis of weighted mean difference of weight loss (kg) in comparison arms that implemented a severely energy-restricted diet of 4 weeks or ≥6 weeks in duration. Abbreviations: ERD = energy-restricted diet; WMD = weighted mean difference; CI = confidence interval; D+L = Laird and DerSemonian method; I-V = inverse variance method.

**Figure 3 behavsci-09-00144-f003:**
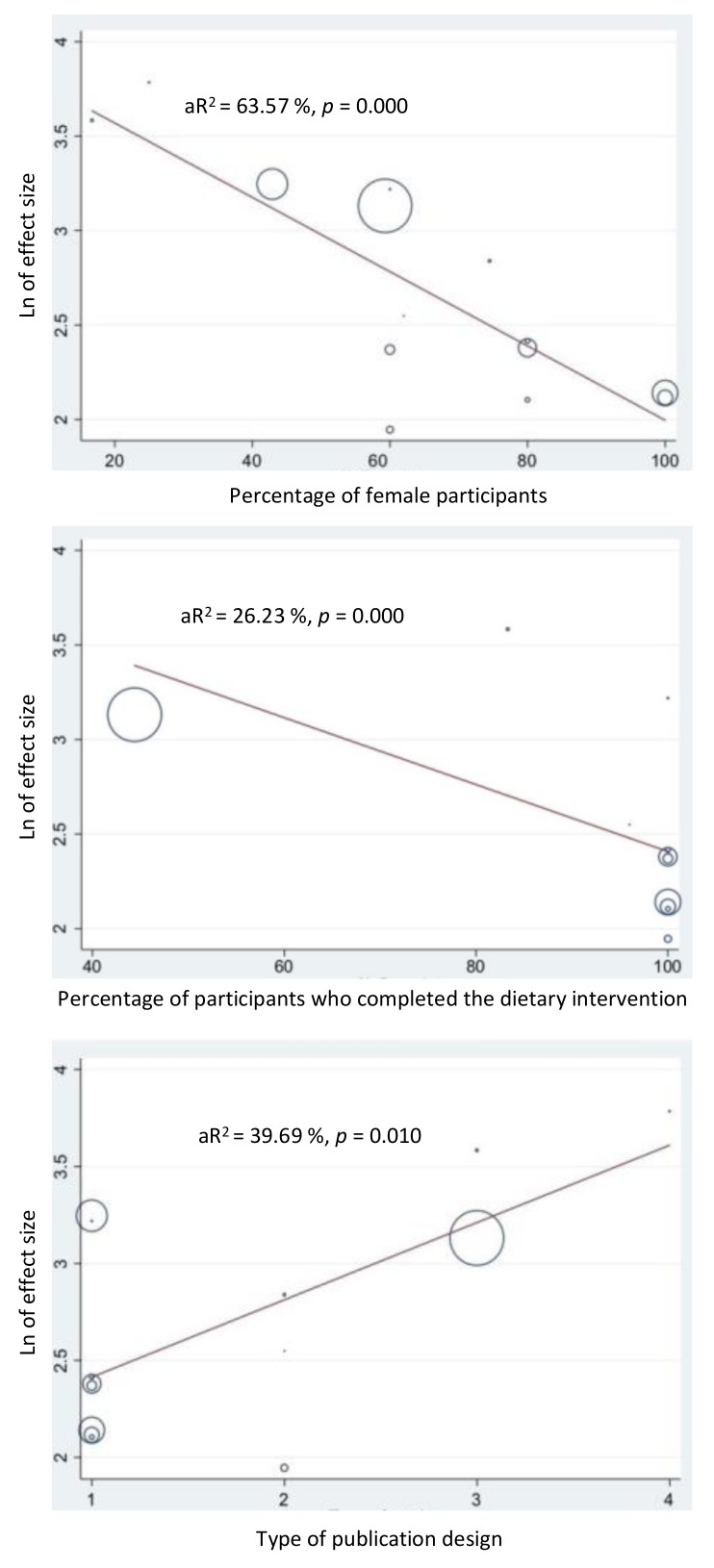
Univariate meta-regression of factors affecting heterogeneity. Abbreviations: Ln = natural logarithm; aR^2^ = adjusted R^2^ statistic.

**Table 1 behavsci-09-00144-t001:** Risk of bias for publications included in this systematic review. * = publication excluded from meta-analysis. Abbreviations: L = low risk (green), M = moderate risk (yellow), H = high risk (red).

Publication First Author and Year of Publication (Reference)	Bias Due to Confounding	Bias in Selection of Participants for the Intervention	Bias in Classification of Interventions	Bias Due to Deviations from Intended interventions	Bias Due to Missing Data	Bias in Measurement of Outcomes	Bias in Selection of the Reported Result	Overall Bias
Contaldo 1981 [58]	M	M	L	L	L	M	L	M
Leonetti 2015 [61]	M	M	M	L	M	M	L	M
Martin 1995 [70]	M	M	M	L	L	M	M	M
Vazquez 1992 [59]	M	M	M	L	L	M	L	M
Hakala 1996 [64]	M	M	M	L	L	M	L	M
Iacobellis 2008 [69]	M	M	L	L	L	M	L	M
Rasmussen 2008 [71]	M	M	M	L	M	M	L	M
Pekkarinen 1997 [60]	M	M	L	L	M	M	L	M
Alabdali 2013 [68]	H	H	H	M	L	M	M	H
Ryan 2010 [72]	L	L	L	L	L	L	L	L
Winkler 2013 [73] *	M	L	L	L	M	M	L	M

**Table 2 behavsci-09-00144-t002:** Characteristics of the severely energy-restricted diets and weight loss outcomes from the 11 publications (with 16 comparison arms) included in this systematic review. Abbreviations: F = female, M = male, VLED = very low energy diet, LED = low energy diet, CHO = carbohydrate, Pro = protein, TMR = total meal replacement diet, FB = food-based diet, NR = not reported, SD = standard deviation, * not included in the meta-analysis.

Publication First Author and Year of Publication (Reference) Comparison Arm Publication Type	Mean Age of Participants at Baseline (Years)	Number of Participants at Recruitment (Sex Breakdown at Recruitment)	Mean Baseline Weight (kg) ± SD, unless Stated Otherwise	Mean Baseline BMI (kg/m^2^) ± SD, unless Stated Otherwise	VLED or LED and Daily Prescription for Energy, Carbohydrate, Fat and Protein Basis of Diet (TMR or FB) Adjunct Therapy	Intervention Duration (Weeks)	Reporting Time from Start of Intervention (Weeks)	Number of Completers (%)			Mean Weight Loss (kg) ± SD, unless Stated Otherwise	Mean Weight Loss (% of Initial Weight) ± SD, unless Stated Otherwise
Contaldo 1981 [58] Low protein Prospective case series	41 ± 2	10 (6 F, 4 M)	NR	44.7 ± 2.6	VLED 330 kJ (80 kcal), 0 g CHO, 0 g Fat, 17 g Pro TMR (with allowance for unlimited tea consumption) No adjunct therapy, inpatient setting	4	4	10 (100%)			10.7 ± 0.8	9.6 ± NR
Contaldo 1981 [58] High protein Prospective case series	43 ± 6	5 (4 F, 1 M)	NR	48.8 ± 2.6	VLED 740 kJ (180 kcal), 0 g CHO, 2 g Fat, 40 g Pro TMR (with allowance for unlimited tea consumption) No adjunct therapy, inpatient setting	4	4	5 (100%)			8.2 ± 0.9	7.0 ± NR
Contaldo 1981 [58] Low to high protein Prospective case series	42 ± 5	5 (4 F, 1 M)	NR	49.0 ± 2.1	2 weeks VLED 330 kJ (80 kcal), 0 g CHO, 0 g Fat, 17 g Pro 2 weeks VLED 740 kJ (180 kcal), 0 g CHO, 2 g Fat, 40 g ProTMR (with allowance for unlimited black tea consumption) No adjunct therapy, inpatient setting	4	4	5 (100%)			10.8 ± 0.3	8.3 ± NR
Contaldo 1981 [58] Low to high protein + CHO Prospective case series	37 ± 2	5 (4 F, 1 M)	NR	43.1 ± 4.4	2 weeks VLED 330 kJ (80 kcal), 0 g CHO, 0 g Fat, 17 g Pro 2 weeks VLED 740 kJ (180 kcal), 25.5 g CHO, 0 g Fat, 17 g Pro TMR (with allowance for unlimited black tea consumption) No adjunct therapy, inpatient setting	4	4	5 (100%)			11.2 ± 1.2	10.5 ± NR
Leonetti 2015 [61] VLED Prospective cohort	48 ± 11	50 (31 F, 19 M)	150.4 ± 26.3	53.5 ± 8.4	Energy ramping protocol: i. 10 days VLED 2450 kJ (586 kcal), 15 g CHO, 24 g Fat, 80 g Pro (i) TMR (with supplemental ketone powder and low starchy vegetables) ii. 10 days VLED 3337 kJ (798 kcal) 55 g CHO, 30 g Fat, 80 g Pro iii. 10 days LED 4631 kJ (1106 kcal), 14.5 g CHO, 33 g Fat, 60 g Pro (ii & iii) FB (Unlimited daily allowance vegetables) No adjunct therapy	4	4	48 (96%)			12.8 ± 21.4	8.5 ± NR
Leonetti 2015 [61] LED Prospective cohort	48 ± 11	30 (18 F, 12 M)	153.2 ± 32.4	53.5 ± 8.4	LED 5000 kJ (1200 kcal), 115 g CHO, 42 g Fat, 90 g Pro FB No adjunct therapy	4	30	30 (100%)			7.0 ± 1.2	4.6 ± NR
Martin 1995 [70] Prospective cohort	40 ± 8	47 (35 F, 12 M)	161.2 ± 31.0	58.4 ± 11.6	VLED 1730 kJ (414 kcal), 0 g CHO, 0 g Fat, 70 g Pro TMR Adjunct behavioural therapy for 4 weeks	4	4	47 (100%)			17.1 ± 9.7	10.6 ± NR
Vazquez 1992 [59] Ketogenic VLED Prospective case series	45 ± 4	8 (8 F)	NR	47.0 ± 2.0	VLED (ketogenic) 2448 kJ (594 kcal), 10 g CHO, 38 g Fat, 52 g Pro TMR (with allowance for unlimited black tea and coffee, and 750 mL daily allowance of diet soft drink) No adjunct therapy, inpatient setting	4	4	8 (100%)			8.5 ± 0.3	NR
Vazquez 1992 [59] Non-ketogenic VLED Prospective case series	43 ± 5	8 (8 F)	NR	49.0 ± 4.0	VLED (non-ketogenic) 2430 kJ (590 kcal), 76 g CHO, 10 g Fat, 50 g Pro TMR (with allowance for unlimited black tea and coffee, and 750 mL daily allowance of diet soft drink) No adjunct therapy, inpatient setting	4	4		8 (100%)	8.3 ± 0.5		NR
Hakala 1996 [64] Prospective case series	44 ± 8	7 (3 F, 4 M)	140.4 ± 22.5	46.6 ± 6.3	VLED 2100 kJ (502 kcal), 47% CHO, 1.8% Fat, 35% Pro TMR Adjunct behavioural therapy for 6 weeks	6	16		7 (100%)	25.7 ± 0.5		18.0 ± NR
Iacobellis 2008 [69] Prospective case series	35 ± 10	20 (12 F, 8 M)	154.0 ± NR	45.0 ± 5.0	LED 3700 kJ (885 kcal), macronutrient prescription NR TMR Adjunct behavioural therapy for 12 weeks	12	24		20 (100%)	25.0 ± 10.0		20.0 ± 2.8
Rasmussen 2008 [71] Non-randomised controlled trial	32 ± 2	6 (1 F, 5 M)	126.0 ± 8.0	41.0 ± 1.0	VLED 1600 kJ (383 kcal) macronutrient prescription NR TMR Adjunct behavioural therapy for 12 weeks	12	12		5 (83%)	36.0 ± 7.0		28.6 ± NR
Pekkarinen 1997 [60] Non-randomised controlled trial	42 ± 9	25 (16 F, 11 M)	131.2 ± 17.7	45.3 ± 4.0	VLED 2100 kJ (502 kcal), 65 g CHO, 2 g Fat, 50 g Pro TMR (Unlimited daily allowance of low-starch vegetables) Adjunct behavioural therapy for 16 weeks	7	7		12 (48%)	22.9 ± 0.5		17.5 ± NR
Alabdali 2013 [68] Retrospective case series	60 ± 8	8 (2 F, 6 M)	NR	57.1 ± 8.8	LED 3762 kJ (900 kcal), 67 g CHO, 30 g Fat, 90 g Pro TMR No adjunct therapy	142 ± 43	112		8 (100%)	44.0 ± 15.0		27.0 ± 13.0
Ryan 2010 [72] Randomised controlled trial	47 ± NR	516 (326 F, 190 M)	128.4 (median)	45.6 ± 7.9 (median)	VLED 3720 kJ (890 kcal), 110 g CHO, 15 g Fat, 75 g Pro TMR (with 10 g daily allowance of oil) Adjunct behavioural therapy for 8 weeks, followed by pharmacotherapy for unknown length of time	8	104		119 (39%)	17.2 ± 1.6		8.3 ± 0.8
Winkler 2013 [73] * Non-randomised controlled trial	44 ± 1	115 (76 F, 39 M)	136.3 ± 2.2	46.6 ± 0.5	VLED 3344 kJ (800 kcal), 100 g CHO, 15 g Fat, 70 g Pro TMR (Unlimited daily allowance of low-starch vegetables) Adjunct behavioural therapy for 12 weeks	12	52		115 (100%)	NR		18.3 ± 0.9

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
