# Peer review of "Effectiveness and Characterization of Severely Energy-Restricted Diets in People with Class III Obesity: Systematic Review and Meta-Analysis"

_behavsci, 2019, doi:10.3390/bs9120144_

Round 1
Reviewer 1 Report
I have following comments:
why calories intake was not same for all candidates under study? Author should use candidates with almost same energy intake. Line 44 and 45: Commonly used method for weight management also included exercise. What was the status of glucose level at the starting of study. Glucose level also effect rate of weigh loss. What was the status of insulin level at the starting of study. Glucose level also effect rate of weigh loss. please described the scientific use of this study. Is there any report of fatty acid or cholestrol level of candidates?
Author Response
Why calories intake was not same for all candidates under study? Author should use candidates with almost same energy intake.
Our response: A systematic review and meta-analysis is a statistical analysis that combines the results of multiple scientific studies that address the same question. An extensive electronic data base search is used to identify all possible publications that meet a specific set of inclusion criterion. Identified publications can have similar, but not necessarily exactly the same intervention (e.g. diet therapies with energy restrictions ranging from 30% to 60%), or in other cases can examine completely different interventions against a common outcome (e.g. effects of diet and / or exercise therapies on weight loss). These differences in protocols used are statistically accounted for in the analysis itself.
In this systematic review and meta-analysis our search criteria involved the collation of all interventions published that used a severely energy-restricted diet on the population cohort with class III obesity. Although the 11 identified studies did have varying dietary energy intakes prescribed, they all met the definition of being a severely-energy restricted diet; that is, diets with a 65 to 75% energy restriction relative to energy requirements (Gibson et al., 2016).
In addition, our post analysis meta-regression used statistical methods to determine whether the variability in dietary energy intakes prescribed in each publication intervention was a significant confounding factor in our results. Our statistical analysis showed that this was not the case: the variability in dietary energy intakes prescribed did not affect the results found.
Line 44 and 45: Commonly used method for weight management also included exercise.
Our response: Line 44 and 45 states “commonly used and effective long-term weight management”. We acknowledge that exercise interventions are a commonly-used method for weight management, however we respectfully disagree that exercise is an effective long-term weight management therapy. Two previous meta-analyses have compared the effectiveness of exercise alone against other weight loss treatment modalities and have found exercise alone to be largely ineffective for acute weight loss and long-term weight management (Franz, 2007; Johns, Hartmann-Boyce, Jebb, & Aveyard, 2014).
What was the status of glucose level at the starting of study. Glucose level also effect rate of weight loss. What was the status of insulin level at the starting of study. Glucose level also effect rate of weight loss. Is there any report of fatty acid or cholesterol level of candidates?
Our response: Each of the 11 publications identified for meta-analysis underwent data extraction on various metabolic markers, and information on circulating levels of glucose, insulin, cholesterol markers and fatty acids were flagged for collection. However, only 2 of the 11 publications reported on baseline glucose and cholesterol levels, and none reported on insulin or fatty acid levels. Therefore, there was not enough data found on these specific circulating analytes to provide review of their effect on weight loss during a severely energy-restricted diet.
Please described the scientific use of this study.
Our response: We have added marked up text to the Discussion of our revised manuscript to further explain the relevance of our new findings.
References
Franz, M. J. (2007). Weight-loss outcomes: a systematic review and meta-analysis of weight-loss clinical trials with a minimum 1-year follow-up. J Am Diet Assoc, 107(10), 1755-1767. doi:10.1016/j.jada.2007.07.017
Gibson, A. A., Seimon, R. V., Franklin, J., Markovic, T. P., Byrne, N. M., Manson, E., . . . Sainsbury, A. (2016). Fast versus slow weight loss: development process and rationale behind the dietary interventions for the TEMPO Diet Trial. Obes Sci Pract, 2(2), 162-173. doi:10.1002/osp4.48
Johns, D. J., Hartmann-Boyce, J., Jebb, S. A., & Aveyard, P. (2014). Diet or Exercise Interventions vs Combined Behavioral Weight Management Programs: A Systematic Review and Meta-Analysis of Direct Comparisons. Journal of the Academy of Nutrition and Dietetics, 114(10), 1557-1568. doi:10.1016/j.jand.2014.07.005
Reviewer 2 Report
The authors made a meta-analysis comprising 11 papers. The authors planned, selected and done the study according to the recent recommendations. The paper is overall well written. My question is important to avoid the duplicate population analysis: Have you checked the two arms of the study published by the same author same year? (like Contaldo 4 papers from1981 and Leonetti from 2015). If it is the case you should report duplication in one arm of the analysis.
Author Response
My question is important to avoid the duplicate population analysis: Have you checked the two arms of the study published by the same author same year? (like Contaldo 4 papers from 1981 and Leonetti from 2015). If it is the case you should report duplication in one arm of the analysis.
Our response: The paper “Contaldo 1981” was a single paper that contained 4 differing study arms. Each of the study arms used different study participants and dietary prescriptions. Hence each comparator arm was used in the meta-analysis as a separate comparator, and thus was not a duplication of the population data.
The paper “Leonetti 2015” was a single paper with 2 differing study arms, an intervention and control arm. Both study arms, intervention and control, met our inclusion criteria and were therefore included in the meta-analysis. The intervention and control arms used different study participants, and varied in dietary energy intake prescriptions and dietary intervention lengths. This resulted in the intervention and control study arms being separated according to <4 week and >6 week classification used in our meta-analysis. Thus, this is not a duplication of population analysis.
Information has been added in marked up text to Section 3.3 of our revised manuscript to clarify the lack of data duplication.
Reviewer 3 Report
Manuscript ID: behavsci-628054
Article: Effectiveness and characterization of severely energy-restricted diets in people with extreme obesity: systematic review and meta-analysis.
P2, lines 45/46, contraindicated for some individuals – elaborate as to why? Lines 47 onwards – definition/criteria of VLEDs and kJ ranges – any international guidelines to refer to these diets classification? Lines 58-59 states: UK guidelines suggest that severely energy-restricted diets can be harmful …… - pls. clarify which guidelines being referred to here, is it NICE?
Lines 60 onwards refer to American College of Cardiology view specifically – are there any other contrasting or agreeing to guidelines by other bodies in the US? Society/College of Obesity or Diabetes for instance? What are the specific issues with the VLEDs? Terms VLEDs and severely energy-restricted diets (SERDs) are used across the manuscript – are they used in a similar context or they differ? Please clarify this in the manuscript and perhaps a consistency of terms will be good. Please clarify as tow which guidelines are been used to classify obesity into extreme or severe (Class I, II, III) for the purpose of the manuscript?
Lines 83 onwards – it appears that only Google Scholar was used – a similar search in other related databases (such as Medline, Cochrane, EMBASE, SCOPUS etc) to be done to confirm this. It appears that inclusion criteria was broadened to include uncontrolled study designs as well as dietary interventions with a higher upper limit of energy intake prescription.
In addition, due to the lack of long-term dietary interventions found, classifications of dietary interventions were categorised using shorter dietary intervention durations than initially proposed. This is not clear in the abstract. Citations after duplicates removed reduced from 951 to 572 – how duplicates were removed? 258 records excluded on the title/abstract screen – what were the exclusion criteria for the title/abstract review? The terms, ‘publications’, ‘citations’ and ‘studies’ are often used ambiguously – please clarify the use, be consistent across the manuscript. Please confirm whether the outlier study (142 weeks duration) is included in final conclusions/data in abstracts? Lines 356/357: …. Mean weight loss of 25.78 Kg – how much in % body weight to compare with study 70?
Overall there needs to be caution in making strong conclusions out of the study due to significant risk of bias and lack of RCTs. Authors do acknowledge this on p.5 however, this should while making strong conclusions elsewhere in the manuscript. It would be interesting to know if there are long term interventions in the area – 4-6 weeks is relatively short and weight regain is common on longer terms. In particular, how effective such diets are on longer-term and how effective they are in terms of patient compliance/adherence. It may be important to note that if data screening and filters are performed independently by two researchers or one performed the initial analysis and other only checked?
Author Response
P2, lines 45/46, contraindicated for some individuals – elaborate as to why?
Our response: We have now mentioned this in the Introduction of our revised manuscript, as shown in marked up text.
Lines 47 onwards – definition/criteria of VLEDs and kJ ranges – any international guidelines to refer to these diets classification?
Our response: This is now also mentioned in marked up text in the Introduction of our revised manuscript.
Lines 58-59 states: UK guidelines suggest that severely energy-restricted diets can be harmful- pls. clarify which guidelines being referred to here, is it NICE?
Our response: Yes, it is the NICE guidelines – this has been added to our revised Introduction section, and is shown in marked up text.
Lines 60 onwards refer to American College of Cardiology view specifically – are there any other contrasting or agreeing to guidelines by other bodies in the US? Society/College of Obesity or Diabetes for instance? What are the specific issues with the VLEDs?
Our response: This has been clarified (in marked up text) in the Introduction section of our revised manuscript.
Terms VLEDs and severely energy-restricted diets (SERDs) are used across the manuscript – are they used in a similar context or they differ? Please clarify this in the manuscript and perhaps a consistency of terms will be good.
Our response: We have revised our manuscript to provide clarity around these terms. Please see marked up text in the Introduction section (line 58) and section 2.2.
Please clarify as to which guidelines are been used to classify obesity into extreme or severe (Class I, II, III) for the purpose of the manuscript?
Our response: WHO guidelines were used, and we have revised our manuscript to explicitly state the source of this BMI classification system, as shown in marked up text in the first paragraph of the Introduction section of the revised manuscript. In addition, we have changed the title of our manuscript to provide consistency in the classification of obesity used.
Lines 83 onwards – it appears that only Google Scholar was used – a similar search in other related databases (such as Medline, Cochrane, EMBASE, SCOPUS etc) to be done to confirm this.
Our response: A search has now been performed in the Cochrane and Embase databases, and the manuscript has been updated accordingly, as shown in marked up text in the last Paragraph of the Introduction section.
It appears that inclusion criteria was broadened to include uncontrolled study designs as well as dietary interventions with a higher upper limit of energy intake prescription.
Our response: We have now mentioned this in the Abstract of our revised manuscript, as well as in Section 2.2 of the Materials and Methods section, as shown in marked up text.
Due to the lack of long-term dietary interventions found, classifications of dietary interventions were categorised using shorter dietary intervention durations than initially proposed. This is not clear in the abstract.
Our response: We have now mentioned this in the Abstract of our revised manuscript, as shown in marked up text.
Citations after duplicates removed reduced from 951 to 572 – how duplicates were removed?
Our response: Additional information has been added to the Materials and Methods of our revised manuscript to explain how duplicates were removed, as shown in marked up text in Section 3.1.
258 records excluded on the title/abstract screen – what were the exclusion criteria for the title/abstract review?
Our response: Additional information to provide transparency around the exclusion of publications during the title/abstract review has been added to Figure 1.
The terms, ‘publications’, ‘citations’ and ‘studies’ are often used ambiguously – please clarify the use, be consistent across the manuscript.
Our response: We have revised our manuscript to provide consistency around these terms, thank you for bringing this to our attention.
Please confirm whether the outlier study (142 weeks duration) is included in final conclusions/data in abstracts?
Our response: The outlier of 142 weeks duration was included in our final conclusions as its inclusion or exclusion did not change the outcome of our results, as indicated in the Results section. To provide clarity around this, additional information has been added to Section 3.6 of our revised manuscript, as shown in marked up text.
Lines 356/357: …. Mean weight loss of 25.78 Kg – how much in % body weight to compare with study 70?
Our response: % body weight loss has now been added to the second last sentence in section 3.8 our revised manuscript, as shown in marked up text.
Overall there needs to be caution in making strong conclusions out of the study due to significant risk of bias and lack of RCTs. Authors do acknowledge this on p.5 however, this should while making strong conclusions elsewhere in the manuscript.
Our response: We have now couched all of the conclusions in our revised manuscript within the context of limitations of the underlying publications, thank you for this feedback.
It would be interesting to know if there are long term interventions in the area – 4-6 weeks is relatively short and weight regain is common on longer terms. In particular, how effective such diets are on longer-term and how effective they are in terms of patient compliance/adherence.
Our response: Additional information on the topic areas of long-term weight management and dietary compliance has been added throughout the Discussion section of our revised manuscript, as shown in marked up text.
It may be important to note that if data screening and filters are performed independently by two researchers or one performed the initial analysis and other only checked?
Our response: Yes, the data screening and filters were performed independently by two researchers. Additional information has been added to the Materials and Methods section 2.3 of our revised manuscript to specify this, as shown in marked up text.
Round 2
Reviewer 1 Report
no